# Exploring health literacy pertaining to general wellbeing and chronic disease management among population registered within Primary Healthcare System: A Study protocol

**Muslim Abbas Syed**[1]*, **Ahmed Sameer Alnuaimi**[2], **Mohamed Ahmed Syed**[3]

**1** Department of Clinical Research, Consultant (Research), Clinical Affairs, Primary Health Care Cooperation Qatar, Doha, Qatar, **2** Department of Clinical Researc, Public Health Research Consultant, Clinical Affairs, Primary Health Care Cooperation Qatar, Doha, Qatar, **3** Department of Clinical Research, Acting Director of Clinical Research, Clinical Affairs, Primary Health Care Cooperation Qatar, Doha, Qatar

* masyed@phcc.gov.qa

## Abstract

### Background

Evidence suggests that high level of health literacy among patients is associated with high levels of accessibility to healthcare, better understanding consent-to-treat forms, increased compliance to treatment (particularly in taking medication for longstanding chronic diseases), comprehending general healthcare and improved preventive care and early disease detection, proper use of home medical devices from patients and increased capability of accessing the various channels of health information when required. The proposed study aims to assess and capture preferences and perceptions regarding health literacy of population (registered within primary care clinics in state of Qatar) in context to their general wellbeing (physical and mental health) and management of chronic disease conditions.

### Methods

The study design will be mixed methods (include both quantitative and qualitative elements). The HLQ survey will be sent online (text messages) to the population registered with PHCC and to capture service users' perceptions and preferences pertaining to health literacy channels. The qualitative method will be utilized to gain an in-depth understanding of the various health literacy channels, challenges and barriers pertaining to the implementation of the health literacy strategies from both the healthcare provider and service users' perspective. The qualitative analysis will be interpreted utilizing the Socioecological Model (SEM). The REPORT statement will be followed during analysis and writing of the study. For the qualitative component we aim to report the results in accordance with 'Consolidation criteria for reporting

**Data availability statement:** No datasets were generated or analysed during the current study. All relevant data from this study will be made available upon study completion.

**Funding:** Qatar National Library funded to cover the publication fees for the study protocol.

**Competing interests:** The authors have declared that no competing interests exist.

qualitative research (COREQ) and 'Standards for reporting qualitative research '(SRQR) guidelines.

## Results

The findings (quantitative and qualitative) will be triangulated to design an evidence-based health literacy framework which can be utilized for service design and re-design to deliver optimal patient centered primary health care services within the country & modelled in similar health care settings geographically.

## Discussion

Evidence generated from the study can increase health literacy levels among service users by modification of health behavior through effective health education. More-over, increased health literacy levels and improved self-care can lead to improved compliance with treatment leading to effective management of diseases, particularly long-term chronic diseases which are associated with multi-morbidity, polypharmacy and challenges associated side effects of prescribed medications.

---

### Section 1: Introduction

Health literacy is defined as 'the ability to read, understand and act on health information' according to the report published by Institute of Medicine in 2004 relating to the significance of the concept in facilitating patients to clear misconceptions regarding their healthcare [1]. Health literacy is further categorized at three levels which encompass functional health literacy which composes basic reading and writing skills, communicative or interactive health literacy which include more advanced cognitive and literacy skills and critical health literacy which is a higher level which enables individuals to achieve more control over decisions regarding their health [2,3].

The concept has recently been widely endorsed by World Health Organization (WHO) and has been recognized as the key indicator in achieving the desired outcomes in health education campaigns aimed towards promoting patient centered care [4–6]. Health literacy levels among patients can have a significant impact in overall accessibility to healthcare services and the ability to comprehend the health information imparted from various sources of the healthcare system [7]. The advantages of high literacy levels among patients include high levels of accessibility to healthcare, more understanding consent-to-treat forms, increased compliance to treatment (particularly in taking medication), comprehending general healthcare and improved preventive care and early disease detection, proper use of home medical devices from patients and increased capability of accessing the various channels of health information when required [1,3].

Moreover, the literature substantiates the fact that individuals with lower health literacy levels are associated with poorer health outcomes in comparison with individuals with higher literacy levels [8–10]. Primary care services are established as the first source of healthcare and play a crucial role in early disease detection, prevention

and controlling the burden of disease (particularly non-communicable diseases) among the population [11]. Therefore, various interventions have been designed to improve health literacy among patients accessing these services to achieve the desired health outcomes [12,13]. It is important to identify health literacy strengths, challenges, and preferences to build locally fit-for-purpose and actions that are pragmatic and can be implemented by active community engagement by local initiatives and ownership [14]. Furthermore, it is worthwhile exploring how the social and cultural determinants of health impact health literacy among patients accessing primary care services. Health literacy can serve as lens for implementing effective treatment modalities pertaining to non-communicable diseases and for better understanding and designing health promotion strategies for general wellbeing specifically within the domains of preventive medicine [15].

The overall aim of the study is to assess and capture preferences and perceptions regarding health literacy of registered population (within primary care clinics in state of Qatar) in context to their general wellbeing (physical and mental health) and management of chronic disease conditions. The study also aims to highlight the various mediums of health education that are currently being utilized and the gaps in practice from a service user's perspective.

The specific objectives of the study include:

1. To assess health literacy levels among service users registered with primary healthcare clinics in Qatar, considering demographic factors such as age, gender, and ethnicity (Qatari and non-Qataris).

2. To explore service users' preferences and healthcare provider views regarding the receipt of health education materials and information, including preferred mediums and pathways, among diverse demographic groups.

3. To assess the message penetration and acceptability of selected communication channels in raising public health awareness among PHCC registered population.

4. To identify strategies and potential gaps in already existing channels of health literacy within PHCC from a service user perspective highlighting the associated intrapersonal, interpersonal, institutional, community and policy level factors.

## Section 2: Methods

### Study settings

'Qatar, a peninsular Arab country (with having world's third-largest natural gas and oil reserves) has recently emerged as a leading primary healthcare provider country that aims to provide comprehensive primary care with due consideration of the fundamental principles outlined within the concept by the World Health Organization. This was mainly achieved by the development of a universal publicly funded primary healthcare service delivered by the PHCC. PHCC is one of the main and most significant primary care providers in the country publicly with 32 health centers. Importantly all the primary health centers are accredited by Accreditation Canada International and distributed across three geographical regions [16].'

### Study design

The study design will be mixed methods (include both quantitative and qualitative elements). The HLQ survey (Arabic and English versions) will be sent online (text messages) to the population registered with PHCC and to capture service users' perceptions and preferences pertaining to health literacy channels. The qualitative method will be utilized to gain an in-depth understanding of the various health literacy channels, challenges and barriers pertaining to the implementation of the health literacy strategies from both the healthcare provider and service users' perspective. This will include interviews with services users accessing primary health care centres and focus group discissions with health care providers (HCPs) who are providing health care services within primary care settings.

## Sampling strategy and recruitment plan for service users registered with PHCC to participate in online HLQ

A full list of adult individuals aged 18 + years registered at Primary Health Care Corporation (PHCC) will be requested from Business Health Intelligence (BHI). The data will be extracted and stratified according to five age bands, two genders and the six major national categories (based on data shared by PHCC's health information management team). The registered population is stratified into 60 strata. To represent the PHCC population a stratified random sample not proportional to size is planned. This sampling technique allows calculating a valid relative frequency estimate for each of the 60 strata individually, while maintaining the representation of the population estimates using a weighting approach based on the sampling fraction. A fixed strata size of 100 is targeted to obtain a total sample size of 6000. Since it is an online administered questionnaire survey the response rate is expected to be as low as 2% [17,18]. The targeted sample size has therefore increased to 300,000 to account for the non-response rate. A sample size of only 2000 will detect a very small difference of 0.1 in the mean score of a specific domain (ranging between 1 and 4 with a standard deviation of 1) between two groups with an estimated Beta power of 0.99 at alpha of 0.05. Therefore, the targeted sample size of 6000 will detect virtually any difference or effect. The respondents will be sent reminder text messages after two weeks to increase the response rate. Regarding the potential on-response bias, the sampling method used will enable valid subgroup analysis by ensuring an adequate sample size in each subgroup. However, any calculated summary measure for the population parameter will be biased by deviations from the sampling frame. This type of bias is addressed by using appropriate subgroup weights, Table 1. Sensitivity analysis (a type of quantitative bias analysis) may be considered to measure the magnitude of sample deviation from the planned sample. *The identifying information (names phone numbers and HC numbers) of the targeted sample will be extracted by PHCC BHI department. The short message service (SMS) will be sent with an approved survey invitation text to the targeted sample. A reminder will be sent after a week. The invitation text message will contain a link to the online questionnaire form using Microsoft office Forms.*

The HLQ consists of 9 scales derived from 44 items rated on either a 4 (scales 1–5; 1 strongly disagree – 4 strongly agree) or a 5-point Likert scale (scales 6–9; 1 cannot do or always difficult – 5 always easy) [19]. The overall HLQ has a Cronbach's alpha of 0.8, demonstrating good internal consistency. [19]The five domains encompassing section 1 of the HLQ scale will be calculated by addition of individual item scores that comprise each domain divided by the total count of those items, as depicted in Table 1 [20]. The scale utilized for section 1 ranged between 1 for "strongly disagree", 2 for "Disagree", 3 for "Agree" and 4 for "Strongly agree". The score for each domain ranges between a minimum of 1 and a maximum of 4. Hence, a high level of the construct, measured in each HLQ domain is verified by having a score >3 as demonstrated in Table 2. The four domains included in section 2 of the HLQ scale will be calculated similarly as demonstrated in Table 2.

### Data analysis plan for the HLQ questionnaire

**Descriptive statistics.** Descriptive statistics, frequencies, percentages, means and standard deviations, will be used to describe the sample (service users who will complete the online HLQ questionnaire) and the HLQ scores (Tables 3–4).

**Interferential statistics.** Interferential statistics, including the Mann-Whitney U Test and the Kruskal-Wallis Test, will be used to explore differences between demographics and HLQ scales [21]. A *p*-value < 0.05 will indicate statistical significance as depicted in Table 5.

### Recruitment strategy, data collection & study tool for focus group discussion among HCPs

A focus group discussion (FGD) will be conducted in which multidisciplinary healthcare professionals will be recruited who are directly involved in providing care to service users accessing primary health centers and have a minimum experience of one year of working in various primary health care corporation affiliated primary health centers. To ensure the sample is representative a diverse cohort of HCPs will be recruited which will mainly include Radiologists, Physicians, Psychiatrist, Dietitians, Health Educators, Nurses, Lab Technologist & Pharmacists (including clinical pharmacist). The HCPs will be recruited by non-probability convenience sampling. The HCPs from the various primary health care centers will be

Table 1. Sampling frame of the registered population at PHCC.

| Age group (years) | Nationality groups | | | | | | | | | | | | | |
|---|---|---|---|---|---|---|---|---|---|---|---|---|---|---|
| | Qatari | | Northern Africa | | South-eastern Asia | | Southern Asia | | Western Asia | | All other | | Total | |
| | N | Sampling fraction (%) | N | Sampling fraction (%) | N | Sampling fraction (%) | N | Sampling fraction (%) | N | Sampling fraction (%) | N | Sampling fraction (%) | N | Sampling fraction (%) |
| **Female** | | | | | | | | | | | | | | |
| 18-29 | 34,922 | 2.8 | 17,584 | 1.4 | 16,943 | 1.4 | 35,222 | 2.8 | 23,781 | 1.9 | 15,043 | 1.2 | 143,495 | 11.6 |
| 30-39 | 24,493 | 2.0 | 26,367 | 2.1 | 64,252 | 5.2 | 57,187 | 4.6 | 26,086 | 2.1 | 23,798 | 1.9 | 222,183 | 17.9 |
| 40-49 | 18,231 | 1.5 | 14,767 | 1.2 | 46,484 | 3.7 | 33,253 | 2.7 | 15,751 | 1.3 | 11,666 | 0.9 | 140,152 | 11.3 |
| 50-59 | 13,675 | 1.1 | 5,851 | 0.5 | 11,612 | 0.9 | 13,763 | 1.1 | 7,382 | 0.6 | 4,844 | 0.4 | 57,127 | 4.6 |
| 60+ | 14,175 | 1.1 | 2,725 | 0.2 | 1,937 | 0.2 | 4,777 | 0.4 | 5,944 | 0.5 | 1,526 | 0.1 | 31,084 | 2.5 |
| Total female | 105,496 | 8.5 | 67,294 | 5.4 | 141,228 | 11.4 | 144,202 | 11.6 | 78,944 | 6.4 | 56,877 | 4.6 | 594,041 | 47.9 |
| **Male** | | | | | | | | | | | | | | |
| 18-29 | 35,793 | 2.9 | 15,472 | 1.2 | 3,108 | 0.3 | 49,719 | 4.0 | 20,056 | 1.6 | 6,103 | 0.5 | 130,251 | 10.5 |
| 30-39 | 22,850 | 1.8 | 32,452 | 2.6 | 8,010 | 0.6 | 111,555 | 9.0 | 24,179 | 1.9 | 9,902 | 0.8 | 208,948 | 16.8 |
| 40-49 | 15,667 | 1.3 | 26,341 | 2.1 | 9,540 | 0.8 | 86,538 | 7.0 | 21,888 | 1.8 | 10,603 | 0.9 | 170,577 | 13.7 |
| 50-59 | 11,775 | 0.9 | 13,133 | 1.1 | 4,432 | 0.4 | 42,783 | 3.4 | 10,708 | 0.9 | 6,992 | 0.6 | 89,823 | 7.2 |
| 60+ | 11,661 | 0.9 | 7,547 | 0.6 | 945 | 0.1 | 15,639 | 1.3 | 8,396 | 0.7 | 3,151 | 0.3 | 47,339 | 3.8 |
| Total male | 97,746 | 7.9 | 94,945 | 7.7 | 26,035 | 2.1 | 306,234 | 24.7 | 85,227 | 6.9 | 36,751 | 3.0 | 646,938 | 52.1 |

**Table 2. HLQ domain description and item order.**

| HLQ domain description | HLQ domain ID | Item order in HLQ questionnaire |
|---|---|---|
| **HLQ Part 1** | | |
| Feeling understood and supported by healthcare providers (4 items) | 1 | 2, 8, 17, 22 |
| High level of the construct: Have an established relationship with at least one healthcare provider who knows them well and who they trust to provide useful advice and information and to assist them to understand information and make decisions about their health | | |
| Having sufficient information to manage my health (4 items) | 2 | 1, 10, 14, 23 |
| High level of the construct: Feel confident that they have all the information that they need to live with and manage their condition and to make decisions | | |
| Actively managing my health (5 items) | 3 | 6, 9, 13, 18, 21 |
| High level of the construct: Recognize the importance and are able to take responsibility for their own health. They proactively engage in their own care and make their own decisions about their health. They make health a priority | | |
| Social support for health (5 items) | 4 | 3, 5, 11, 15, 19 |
| High level of the construct: A person's social system provides them with all the support they want or need for health | | |
| Appraisal of health information (5 items) | 5 | 4, 7, 12, 16, 20 |
| High level of the construct: Able to identify good information and reliable sources of information. They can resolve conflicting information by themselves or with help from others | | |
| **HLQ part 2** | | |
| Ability to actively engage with healthcare providers (5 items) | 6 | 2, 4, 7, 15, 20 |
| High level of the construct: Are proactive about their health and feel in control in relationships with healthcare providers. Are able to seek advice from additional healthcare providers when necessary. They keep going until they get what they want. Empowered | | |
| Navigating the healthcare system (6 items) | 7 | 1, 8, 11, 13, 16, 19 |
| High level of the construct: Able to find out about services and supports so they get all their needs met. Able to advocate on their own behalf at the system and service level | | |
| Ability to find good health information (5 items) | 8 | 3, 6, 10, 14, 18 |
| High level of the construct: Are 'information explorer'. Actively use a diverse range of sources to find information and are up to date | | |
| Understanding health information well enough to know what to do (5 items) | 9 | 5, 9, 12, 17, 21 |
| High level of the construct: Are able to understand all written information (including numerical information) in relation to their health and able to write appropriately on forms where required | | |

included in the study by approaching the managers of the primary health centers who will help identify and provide their contact details for recruitment. The HCPs once identified will be provided with participant information leaflet and copy of FGD guide for a better understanding of the purpose of the study and the themes to be discussed during the discussion. The HCPs will be requested to further help identify other colleagues by snowballing technique. Prior to starting the FGD the participants will be briefly informed about the purpose of the study and key themes included in the discussion. The participants will be reminded that their participation in the FGD is voluntary, and they have the right to end the discussion at any point they feel uncomfortable or decide to no longer participate. The verbal consent will be recorded and a log with starting time and date will be entered in Microsoft Excel sheet. The FGD will be audio recorded, and notes will also be taken during the session utilizing an FGD guide. The FGD will last for approximately 75 minutes. It will be ensured by the principal investigator that there is active participation by HCPs from different specialties while conducting the FGD by referring to each participant while discussing a specific theme. The main themes of the topic guide (supplementary

**Table 3. Demographic details of the participants.**

| Characteristic | n | % |
|---|---|---|
| Age (Mean- years, SD & range) | | |
| Gender (Mean- years, SD & range) | | |
| Ethnic group (Mean- years, SD & range) | | |
| Education (Mean- years, SD & range) | | |
| Relationship status (Mean- years, SD & range) | | |
| Total household income (Mean- years, SD & range) | | |
| Current employment status (Mean- years, SD & range) | | |
| Years living in Qatar (Mean- years, SD & range) | | |
| Diagnosed with a disease condition (Mean- years, SD & range). | | |
| Diabetes Mellitus. | | |
| Chronic kidney disease. | | |
| Musculoskeletal (arthritis, osteoporosis, back pain or other). | | |
| Depression, anxiety or other mental health conditions. | | |
| Heart disease. | | |
| Asthma, emphysema or other respiratory conditions. | | |
| Cancer. | | |
| Stroke, multiple sclerosis or other neurological condition. | | |

document (S1 file)) utilized for the focus group discussions were derived from the broader themes outlined literature and relevant published literature pertaining to the various domains of health literacy, associated channels and existing challenges and gaps in its implementation within primary care settings [22–27].

## Recruitment strategy, data collection & study tool for interview with service users accessing primary health care centers

The healthcare managers of the primary health centers affiliated with Primary Health Care Corporation will be contacted via email to obtain permission to approach service uses by accessing the service to participate in face-to-face interviews while they wait for their consultation in the waiting area of the health center. A separate room will be requested to be allocated by the health manager of the primary health center to conduct the interviews. The service users will be approached by the principal investigator (MA) and invited to participate in the study. The interviews will be conducted by PI. The PI has the relevant academic qualifications and substantial experience of conducting qualitative research and publishing in high impact journals. This is in accordance with the criterion of COREQ and SRQR regarding standards required by the researcher conducting qualitative research. When a service user agrees to participate, they will be invited to the allotted interview room. Prior to starting the interview, the purpose of the study will be explained in detail to the service user, and they will be given the opportunity to ask further questions to ensure that their participation is voluntary, informed and non-coercive. The written consent will be taken by the interviewer before starting the interview. The interviews will be recorded with permission of the service user. Once the interview has taken place all contact information about the service user will be securely destroyed and all the data collected will be anonymised. The interviews are semi-structured and will be conducted utilising an interview guide and script (supplementary document). The interviews will last between 45–60 minutes.

## Analysis of the qualitative data collected from FGD and interviews

The qualitative data once recorded will be transcribed verbatim, and then analyzed using thematic analysis. [28] This approach encompasses 'interpreting, exploring, and reporting patterns and clusters of meaning within the given data' [29]

**Table 4. HLQ scores.**

| HLQ Scale | Mean SD Range ------------ |
|---|---|
| Domain 1: Feeling understood and supported by healthcare providers | |
| Domain 2: Having sufficient information | |
| Domain 3: Actively managing health | |
| Domain 4: Social support for health | |
| Domain 5: Appraisal of health information | |
| Domain 6: Understand health information well enough to know what to do | |
| Domain 7: Active engagement with healthcare providers | |
| Domain 8: Ability to find good health information | |
| Domain 9: Navigating the healthcare system | |

**Table 5. Association between HLQ scores and demographics.**

| Variables | Domain 1 Mean (SE) | Domain 2 Mean (SE) | Domain 3 Mean (SE) | Domain 4 Mean (SE) | Domain 5 Mean (SE) | Domain 6 Mean (SE) | Domain 7 Mean (SE) | Domain 8 Mean (SE) | Domain 9 Mean (SE) |
|---|---|---|---|---|---|---|---|---|---|
| Age p-value | | | | | | | | | |
| Gender p-value | | | | | | | | | |
| Ethnic group p-value | | | | | | | | | |
| Education p-value | | | | | | | | | |
| Relationship status p-value | | | | | | | | | |
| Household income p-value | | | | | | | | | |
| Current employment status p-value | | | | | | | | | |
| Years living in Qatar p-value | | | | | | | | | |

and will be facilitated by reading and re-reading the transcripts for a full familiarization. This will be followed by application of open codes to four transcripts to identify emerging themes of relevance by two researchers (MA AND ASA) [30]. A Computer Assisted Qualitative Data Analysis (CAQDAS) package (NVivo 12 for Windows) will be utilized for this process. This will be followed by agreement by the two researchers (MA and ASA) on a set of codes which will be used with the rest of the transcripts. During this stage categories will be constructed and defined by grouping of codes. This will lead to the development of a working coding framework which will be utilized with the rest of the data and amended as necessary. The study will report the results in accordance with 'Consolidation criteria for reporting qualitative research (COREQ) and 'Standards for reporting qualitative research '(SRQR) guidelines [31,32].

## Interpretation of the qualitative findings

The qualitative analysis will be interpreted utilizing the Socioecological Model (SEM) which will provide to explore the wider determinants of health literacy of service users accessing the primary health care services. SEM is a widely accepted framework for comprehending and describing health determinants and in its entirety can be effectively utilized to explore the factors associated with health literacy of service users nested within the SEM [33]. Although the SEM is widely acknowledged as a framework for understanding wider health determinants, there are only a few examples embedding health literacy in socioecological context [33]. The existing literature only partially explores determinants nested within the SEM and does not investigate the model's entirety when examining factors related to health literacy of service users accessing primary care services. The state of Qatar has a highly developed and extensive primary care health system, Primary Health Care Corporation (PHCC) which is composed of 32 primary health care centers which are scattered throughout the country. PHCC has the prime objective to deliver universal health coverage to its registered population. Each primary health care center is equipped with modern healthcare facilities managed by an international multidisciplinary team of health care professionals delivering the highest standards of primary health care services to a diverse multi-ethnic population (comprising of approximately 88% expats) [16,34]. However, studies report that despite the strong primary health care infrastructure, Qatar has a high prevalence of non-communicable disease and challenges associated with uptake of the services among the population registered with PHCC with diverse ethnic backgrounds [34,35] The model will be populated by the important themes emerging from the qualitative interviews with the patients and FGDs with HCWs to highlight the intrapersonal, interpersonal, Institutional, Community and Policy level factors [Fig 1].

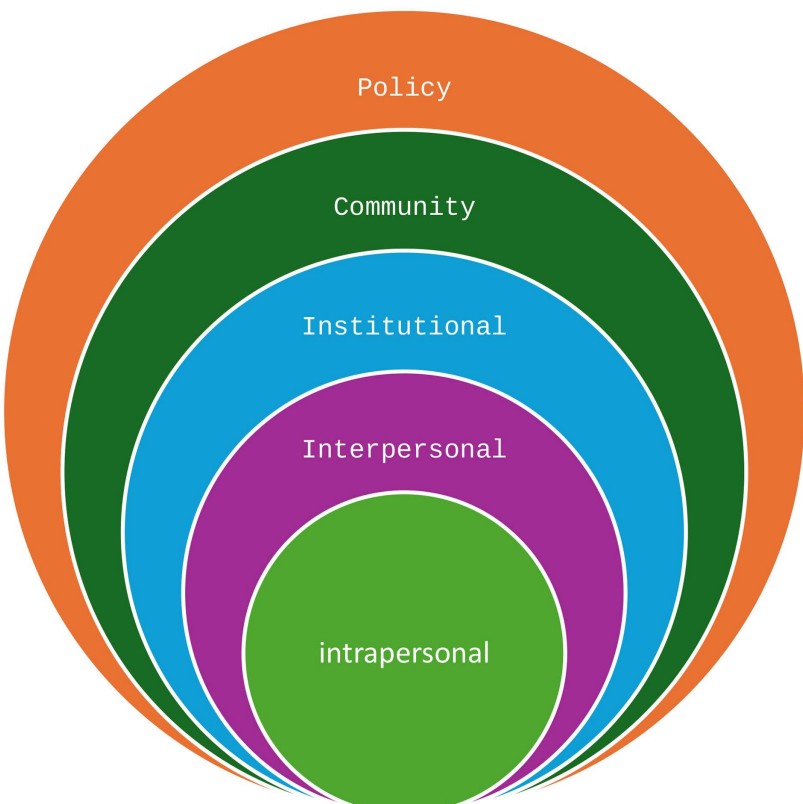

**Fig 1. The SEM categories.**

## Quality control and good practice measures

The REPORT statement, which is an extension of the STROB statement checklist (international, collaborative initiative of epidemiologists, methodologists, statisticians, researchers and journal editors involved in the conduct and dissemination of observational studies, with the common aim of Strengthening the Reporting of Observational studies in Epidemiology) specially designed to assure the quality of reporting of secondary data analysis will be followed during analysis and writing of the report. For the qualitative component we aim to report the results in accordance with 'Consolidation criteria for reporting qualitative research (COREQ) and 'Standards for reporting qualitative research '(SRQR) guidelines [31,32].

   To ensure the ethical and regulatory integrity of the study, oversight mechanisms will be implemented throughout its duration. A designated research monitor from the Clinical Research Department will conduct periodic monitoring and auditing activities. These will verify that the study is being conducted in accordance with the protocol approved by the Institutional Review Board (IRB), and in compliance with Good Clinical Practice (GCP) guidelines and Policies, Regulations and Guidelines for Research Involving Human as outlined by Ministry of Public Health (MoPH) Qatar. Monitoring will include the identification and documentation of any protocol deviations or instances of non-compliance. All findings will be addressed promptly to uphold participant safety and maintain the scientific validity of the research.

## Timelines for study

The study is expected to be initiated in December 2024. The tentative timelines for the main study tasks are summarized in the table below:

| Components of the research study | Expected completion dates |
|---|---|
| Participant recruitment | Dec 2024- Feb 2025 |
| Data collection | Feb 2025- Oct 2025 |
| Drafting of results | Nov 2025-Dec 2025 |

## Ethical considerations

Written consent will be obtained by the principal investigator (MA) prior to the start of the interviews with service users and will be recorded on the digital audio recorder with the participants' permission. At all stages of the interview, it will be emphasised that participation is voluntary, and that the data generated will be anonymous. It will be ensured and made clear to the participants that they could withhold any information that they feel is too sensitive or withdraw from the study at any time. The interviews will be transcribed and anonymised soon after the interviews and the digital recordings of the interviews will be removed from the digital devices. The anonymised transcripts and digital recordings will be stored securely on a server within the clinical research department of Primary Health Care Corporation. Ethics approval was received to conduct the study from Institutional Review Board of Primary Health Care Corporation (supplementary document).

## Patient and public involvement

There was no Patient and Public Involvement (PPI) while formulating the study tools and designing the research methodology. However, for future research and further exploration in designing innovative interventions to improve the health literacy of service users accessing primary health care services there will be active involvement of PPI in pilot testing and designing such interventions.

## Dissemination plan of study findings

Upon completion of the study, findings will be disseminated through multiple channels to ensure broad and meaningful impact. Results will be submitted for publication in peer-reviewed journals and presented at relevant scientific conferences.

In addition, tailored summaries will be shared with key stakeholders, including healthcare providers and policymakers, to inform practice and decision-making. Where appropriate, public-facing materials will be developed to communicate findings to the public, thereby enhancing transparency and promoting community engagement with the research outcomes.

## Section 3: Discussion

The mixed methods study aims to assess the existing health literacy channels operating within the primary care health system and to identify the existing gaps and challenges from key stakeholders' perspectives including both the services users registered with PHCC and the health care professionals. The key objective of the study also includes determining the health literacy levels of service users at population level. The findings of the HLQ highlight health literacy levels at population level and its interrelationship with the sociodemographic indicators whereas the in-depth qualitative investigation highlights the various factors nested within the SEM in context to health literacy of service users accessing primary care services and identifying the potential gaps and challenges. These findings will be triangulated to design an evidence-based health literacy framework which can be utilized for service design and re-design to deliver optimal patient centered primary health care services within the country & modelled in similar health care settings geographically. The immediate and long-term outcomes of the research are outlined in Fig 2.

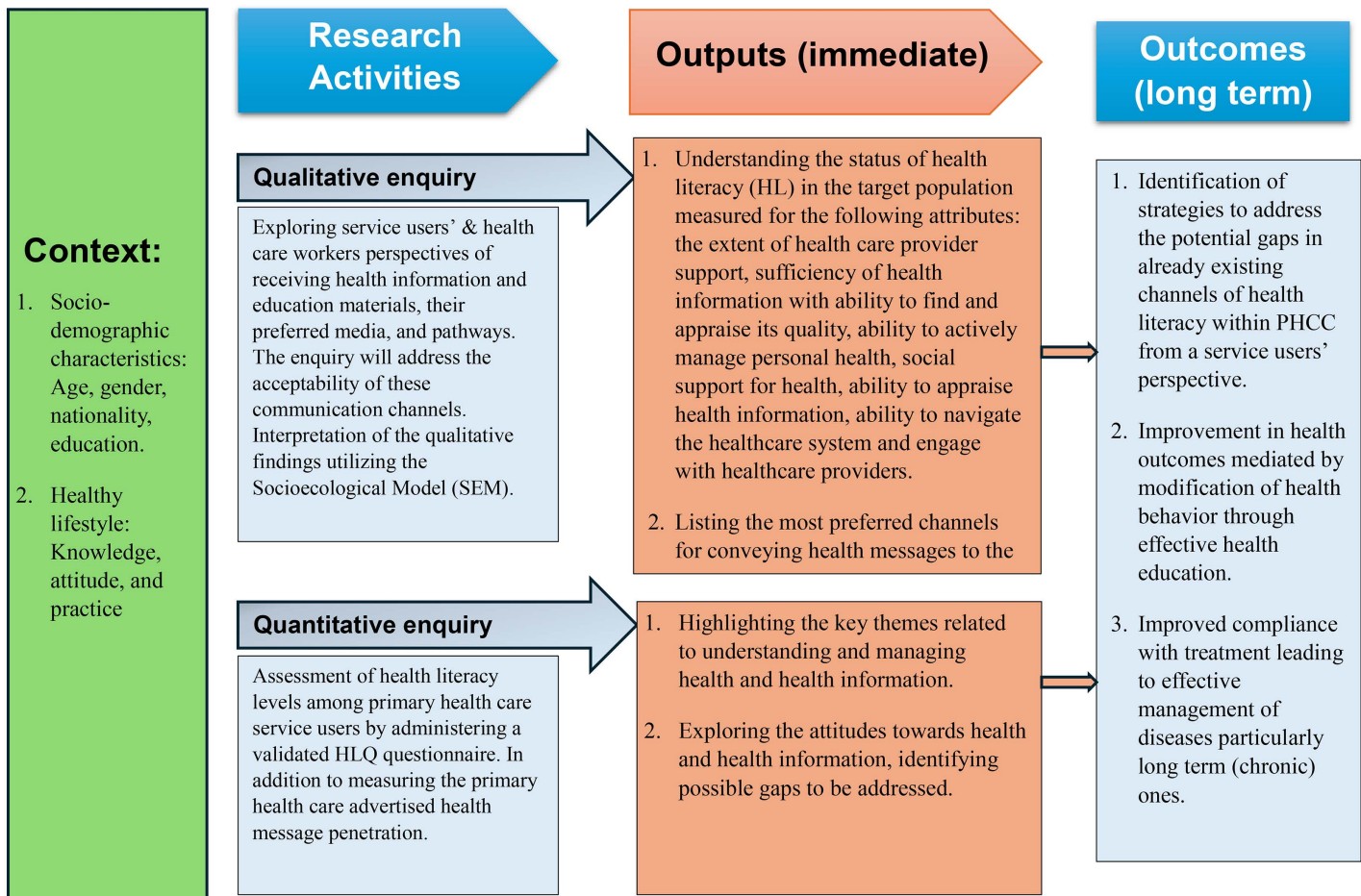

**Fig 2. Conceptual framework of the context, research methodology and immediate and long-term outcomes of the proposed study.**

The conceptual framework of the proposed study regarding the context, research methodology and short and long-terms outcomes expected of the proposed study is depicted in Fig 2. The interrelationship between the levels of health literacy [2,3], the gaps and challenges in implementing health literacy interventions, the proposed strategies to upscale and improve the existing channels and its impact on overall health literacy levels (the nine essential domains of health literacy encompassed in HLQ [20]) is illustrated in Fig 3.

Evidence suggests that high level of health literacy among patients is associated with high levels of accessibility to healthcare, better understanding consent-to-treat forms, increased compliance to treatment (particularly in taking medication for longstanding chronic diseases), comprehending general healthcare and improved preventive care and early disease detection, proper use of home medical devices from patients and increased capability of accessing the various channels of health information when required [36–42]. On the contrary, a study conducted in Australia in acute public hospital settings reported no association between lower health literacy and greater use of hospital health services [37]. However, it was documented in the study that increased age, non-English speaking at home, culturally and linguistically diverse background were all associated with having the most health literacy challenges. [37] Moreover, increased health literacy levels and improved self-care can lead to improved compliance with treatment leading to effective management of diseases particularly long-term chronic diseases which are associated with multi-morbidity, polypharmacy and challenges associated side effects of prescribed medications. [43]

The impact of cultural and linguistic barriers in health literacy and its association with poorer health outcomes are well documented in recent literature [44–46]. Interestingly studies [45,47–50] suggest that native service users have higher rates of accessibility to the primary healthcare services, achieve desired health outcomes and report lesser unmet health care needs in comparison with immigrant population.

Furthermore, literature highlights that interactive health literacy training interventions for health care staff working in primary care settings have the potential to improve their knowledge, approach, and confidence in using health literacy strategies with service users and families accessing the service. [51] Evidence also suggests that integrative models of

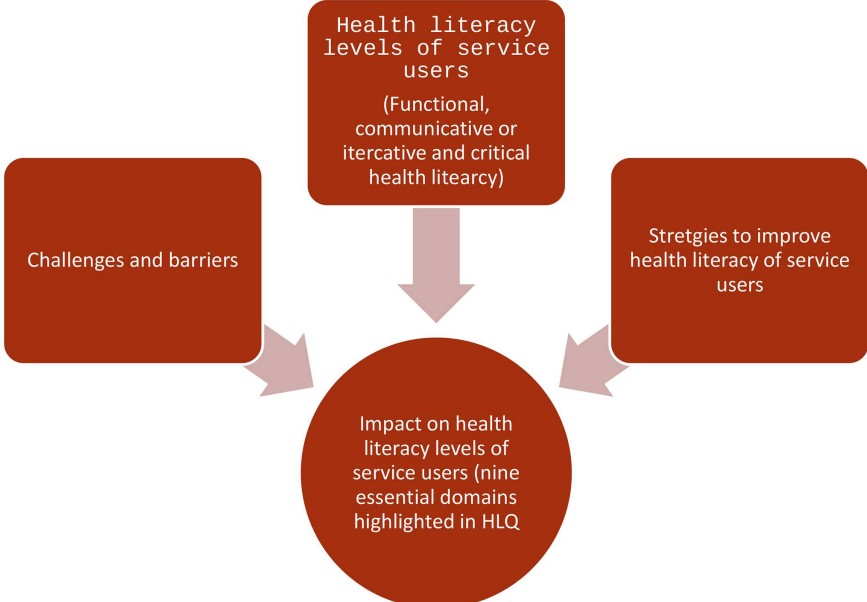

**Fig 3. Interrelationship of health literacy levels, challenges and barriers and strategies to improve health literacy of service users.**

care composed of multidisciplinary teams can increase quality of care, patient satisfaction, health literacy levels and can lead to better chronic disease management. The significance of care giver health literacy is also well document in recent studies which report increased health literacy of care givers can promote efficient utilization of the health care services [52,53].

The limitation of the study includes the probability of a high non-response rate to the online HLQ questionnaire that will be distributed to the registered population with PHCC. Keeping that in view we have considered keeping the response rate to be as low as 2%. The targeted sample size has therefore increased to 300,000 to account for the non-response rate. However, the respondents will be sent reminder text messages after two weeks to increase the response rate. Another limitation could be the non-probability convenience sampling strategy to recruit service users to conduct interviews to capture their perceptions regarding the different health literacy channels. To address this issue, we will conduct interviews in different primary care clinics (out of the 32 clinics spread across the state of Qatar) till saturation of data is achieved (a point where no new concepts, ideas or themes are emerging [54]) and will aim to include equal representation of male and female participants considering their ethnic backgrounds. Moreover, the FGD will include a diverse cohort (multidisciplinary & multiethnic) of HCPs involved in providing primary care in PHCC which will establish that the findings are generalizable, and the sample is representative.

## Supporting Information

**S1 file**
(docx)

## Author contributions

**Conceptualization:** Muslim Abbas Syed, Ahmed Sameer Alnuaimi, Mohamed Ahmed Syed.

**Data curation:** Muslim Abbas Syed, Mohamed Ahmed Syed.

**Formal analysis:** Muslim Abbas Syed.

**Funding acquisition:** Muslim Abbas Syed.

**Investigation:** Muslim Abbas Syed.

**Methodology:** Muslim Abbas Syed.

**Project administration:** Muslim Abbas Syed, Ahmed Sameer Alnuaimi, Mohamed Ahmed Syed.

**Resources:** Muslim Abbas Syed.

**Software:** Muslim Abbas Syed.

**Supervision:** Muslim Abbas Syed, Mohamed Ahmed Syed.

**Validation:** Muslim Abbas Syed, Mohamed Ahmed Syed.

**Visualization:** Muslim Abbas Syed.

**Writing – original draft:** Muslim Abbas Syed, Ahmed Sameer Alnuaimi, Mohamed Ahmed Syed.

**Writing – review & editing:** Muslim Abbas Syed, Ahmed Sameer Alnuaimi, Mohamed Ahmed Syed.

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
