## [Decision Letter · Decision Letter 0]

4 Jun 2025

PONE-D-24-46029Exploring health literacy pertaining to general wellbeing and chronic disease management among population registered within Primary Healthcare System: A Study protocol.PLOS ONE

Dear Dr. Syed,

Thank you for submitting your manuscript to PLOS ONE. After careful consideration, we feel that it has merit but does not fully meet PLOS ONE’s publication criteria as it currently stands. Therefore, we invite you to submit a revised version of the manuscript that addresses the points raised during the review process.

After carefully reviewing the manuscript and considering the feedback provided by the three reviewers, I find that the manuscript has potential; however, it currently falls short of the standards required for publication in its present form. The reviewers have raised several important concerns, particularly regarding methodological rigor, which must be thoroughly addressed. Given the **substantive nature of the revisions required** , I am recommending a **major revision** .

We look forward to receiving your revised manuscript.

Kind regards,

Hansani Madushika Abeywickrama, Ph.D.

Academic Editor

PLOS ONE

Journal Requirements:

Reviewers' comments:

Reviewer's Responses to Questions

**Comments to the Author**

1. Does the manuscript provide a valid rationale for the proposed study, with clearly identified and justified research questions?

Reviewer #1: No

Reviewer #2: Yes

Reviewer #3: Yes

2. Is the protocol technically sound and planned in a manner that will lead to a meaningful outcome and allow testing the stated hypotheses?

Reviewer #1: No

Reviewer #2: Yes

Reviewer #3: Yes

3. Is the methodology feasible and described in sufficient detail to allow the work to be replicable?

Reviewer #1: No

Reviewer #2: Yes

Reviewer #3: Yes

4. Have the authors described where all data underlying the findings will be made available when the study is complete?

Reviewer #1: No

Reviewer #2: No

Reviewer #3: Yes

5. Is the manuscript presented in an intelligible fashion and written in standard English?

Reviewer #1: No

Reviewer #2: Yes

Reviewer #3: Yes

6. Review Comments to the Author

You may also provide optional suggestions and comments to authors that they might find helpful in planning their study.

Reviewer #1: Dear authors,

This manuscript presents a protocol for a national survey to assess health literacy in primary care in Qatar. The manuscript lacks clarity and an evidence-base in the methods, with no mention of the process of developing the data collection tools (i.e., survey and questions, plus interview guide), the process of recruitment and gaining consent from participants, and the planned and detailed data analysis (including methods of integration quantitative and qualitative datasets).

In addition, there is no patient and public involvement and engagement within the protocol methods, nor is there reflection and plans for involving public and patients in the Discussion. The Methods lacks supporting references throughout, particularly in terms of survey development, and the sample size estimation is a gross estimation.

I cannot support the manuscript for consideration for publication as a protocol in PLOS One based on this current version.

Yours sincerely,

James

Reviewer #2: This protocol addresses a timely and highly relevant public health concern—health literacy in chronic disease prevention and management within the primary care setting. The study’s design, ethics compliance, and clarity of objectives demonstrate strong potential to contribute valuable insights to health systems in Qatar and similar contexts. The submission is generally well-prepared for publication, pending minor clarifications.

Strengths

1. Relevance and Timeliness

The focus on health literacy in the context of chronic disease management directly aligns with national and global priorities, particularly WHO health promotion strategies.

2. Rigorous Sampling Strategy

The use of stratified random sampling from PHCC’s registry enhances representativeness across age, gender, and nationality, improving generalizability.

3. Validated Instrument

Use of the HLQ in both English and Arabic strengthens the reliability and cultural appropriateness of the tool.

4. Mixed Methods Approach

The integration of both quantitative and qualitative phases will provide a richer understanding of the issue under study.

5. Ethics and Transparency

IRB approval, consent processes, and attached documentation meet international ethical standards.

6. Document Completion

Required materials, including tools, translated forms, and supplementary documents, have been submitted.

Suggestions for Improvement

1. Missing Demographic Data in the Study Tool

The submitted Health Literacy Questionnaire (HLQ) appears to lack demographic questions (e.g., age, gender, education level, nationality). This is a critical omission for the following reasons:

• Descriptive analysis requires participant characteristics to interpret findings meaningfully.

• Subgroup analysis is necessary to identify disparities or variations in health literacy.

• Equity considerations – health literacy is often shaped by socioeconomic and cultural factors; excluding demographics limits the ability to draw meaningful conclusions and policy recommendations.

Recommendation:

Clarify whether demographic data will be obtained separately (e.g., through PHCC registry) or amend the questionnaire to include a brief demographic section. If already planned, this should be stated clearly in the methods section.

2. Participant Recruitment Details

• The assumed 2% response rate should be supported with rationale or precedent from similar PHCC-based studies or surveys in Qatar. This adds credibility to the feasibility of the recruitment plan.

3. Qualitative Sampling

• While non-probability sampling is acceptable in qualitative research, it would be beneficial to mention efforts to ensure diversity (gender, education, cultural background) in interview participants to reduce potential bias.

4. Data Analysis Plan

• Although HLQ scoring methods are referenced, the manuscript would benefit from a concise overview of intended statistical methods, including any subgroup analyses, correlation assessments, or use of regression models.

5. Conceptual Framework (Figure 1)

• The figure is valuable but visually complex. A clearer flow with consistent formatting and labeled sections would improve readability for international readers unfamiliar with Qatar's healthcare system.

6. Language and Style

• Minor grammatical and structural edits are recommended. For example:

o Original: “Improved compliance with treatment leading to effective management…” Suggested: “…which may improve treatment compliance and disease management.”

• Data Availability Statement: While the current version is compliant, consider stating your intention to deposit anonymized data post-study (e.g., in a public repository or institutional archive).

Recommendation: Minor Revisions

The study protocol is methodologically sound, ethically appropriate, and well-aligned with current health system needs. Minor adjustments related to demographic data collection, recruitment rationale, and clarity of analysis plan are needed. Once addressed, this work will make a meaningful contribution to the literature on health literacy and chronic disease management in the region.

Reviewer #3: This protocol addresses a timely and important research topic with methodological rigor. Addressing the minor clarifications below will further strengthen the manuscript and ensure its value to the scientific and public health communities.

Response Rate and Potential Bias:

While the protocol anticipates a very low response rate and compensates by increasing the invitation pool, it may be beneficial to elaborate further on strategies to maximize response rates (e.g., use of reminders, engagement incentives) and to discuss potential non-response bias in greater detail.

Qualitative Sampling:

The use of convenience sampling for qualitative interviews is acknowledged as a limitation. Consider discussing how heterogeneity will be maximized (e.g., purposive selection based on demographic characteristics) to enhance the credibility of qualitative findings.

Patient and Public Involvement:

If applicable, clarify whether patients or members of the public were involved in the development of the research questions, survey tool adaptation, or interview guide, as this is increasingly encouraged in health research.

7. PLOS authors have the option to publish the peer review history of their article (what does this mean? ). If published, this will include your full peer review and any attached files.

**Do you want your identity to be public for this peer review?** For information about this choice, including consent withdrawal, please see our Privacy Policy .

Reviewer #1: **Yes: ** James Gavin

Reviewer #2: No

Reviewer #3: **Yes: ** Prof. Ananda Chandrasekara

---

## [Author Response · Author response to Decision Letter 1]

10 Jun 2025

Reviewer 1

Comment: This manuscript presents a protocol for a national survey to assess health literacy in primary care in Qatar. The manuscript lacks clarity and an evidence-base in the methods, with no mention of the process of developing the data collection tools (i.e., survey and questions, plus interview guide), the process of recruitment and gaining consent from participants, and the planned and detailed data analysis (including methods of integration quantitative and qualitative datasets). In addition, there is no patient and public involvement and engagement within the protocol methods, nor is there reflection and plans for involving public and patients in the Discussion. The Methods lacks supporting references throughout, particularly in terms of survey development, and the sample size estimation is a gross estimation.

Response: Thank you for the important comments. We have carefully considered all the points mentioned and added the suggested details in the methods section with references and further expanded the discussion component and added additional framework (figure 2) for more clarity and detail. Your important and comprehensive feedback has helped us strengthen the protocol and add important details and relevant references supporting the evidence.

As kindly highlighted in your comment we have added details about the study tool (HLQ) domains and scoring, and the interview and focus group guide. While the protocol was under review, we also got the amendment approved to conduct focus group discission with health care providers (HCPs) working in PHCC. The amendment approval letter by institutional review board is also attached in the supplementary documents. This further strengthens the study as qualitative analysis (both from HCPs and service users’ perspective) will highlight important themes pertaining to the existing health literacy channels, gaps and challenges and strategies to further strengthen its implementation to promote patient centered care and further improve the quality of the services. Moreover, the sociodemographic details are also added in the HLQ. The study tools are attached as supplementary documents.

Furthermore, as kindly suggested references are included for both the quantitative and qualitative study tool development and interpretation (track changes). Details are added about the consent process and recruitment of the participants for interviews, FGDs and HLQ (track changes). A section on patient and public involvement is also added in the protocol. Reference for sample size is also provided.

Reviewer 2

This protocol addresses a timely and highly relevant public health concern—health literacy in chronic disease prevention and management within the primary care setting. The study’s design, ethics compliance, and clarity of objectives demonstrate strong potential to contribute valuable insights to health systems in Qatar and similar contexts. The submission is generally well-prepared for publication, pending minor clarifications.

Strengths

1. Relevance and Timeliness

The focus on health literacy in the context of chronic disease management directly aligns with national and global priorities, particularly WHO health promotion strategies.

2. Rigorous Sampling Strategy

The use of stratified random sampling from PHCC’s registry enhances representativeness across age, gender, and nationality, improving generalizability.

3. Validated Instrument

Use of the HLQ in both English and Arabic strengthens the reliability and cultural appropriateness of the tool.

4. Mixed Methods Approach

The integration of both quantitative and qualitative phases will provide a richer understanding of the issue under study.

5. Ethics and Transparency

IRB approval, consent processes, and attached documentation meet international ethical standards.

6. Document Completion

Required materials, including tools, translated forms, and supplementary documents, have been submitted.

Response: Thank you for the positive feedback and meticulous review. We are truly grateful. We have carefully considered all the important suggestions provided which has further strengthened the study protocol. Individual responses are provided below and highlighted in the track changes version of the manuscript.

Reviewer 2: Suggestions for Improvement

1. Missing Demographic Data in the Study Tool

The submitted Health Literacy Questionnaire (HLQ) appears to lack demographic questions (e.g., age, gender, education level, nationality). This is a critical omission for the following reasons:

• Descriptive analysis requires participant characteristics to interpret findings meaningfully.

• Subgroup analysis is necessary to identify disparities or variations in health literacy.

• Equity considerations – health literacy is often shaped by socioeconomic and cultural factors; excluding demographics limits the ability to draw meaningful conclusions and policy recommendations.

Recommendation:

Clarify whether demographic data will be obtained separately (e.g., through PHCC registry) or amend the questionnaire to include a brief demographic section. If already planned, this should be stated clearly in the methods section.

Response: Thank you for the important suggestion. We fully agree with you. The socio-demographic details are now added in the HLQ, and the amended version of the online tool is added as supplementary documents.

2. Participant Recruitment Details

• The assumed 2% response rate should be supported with rationale or precedent from similar PHCC-based studies or surveys in Qatar. This adds credibility to the feasibility of the recruitment plan.

Response: Thank you for the important comment. We have included reference to support the assumed 2% response rate (track changes).

3. Qualitative Sampling

• While non-probability sampling is acceptable in qualitative research, it would be beneficial to mention efforts to ensure diversity (gender, education, cultural background) in interview participants to reduce potential bias.

Response: Thank you for highlighting a very important point. We have added these details in the recruitment strategy for conducting interviews among service users and FGD with healthcare providers. While the protocol was under review, we also got the amendment approved to conduct focus group discission with health care providers (HCPs) working in PHCC. The amendment approval letter by institutional review board is also attached in the supplementary documents. This further strengthens the study as qualitative analysis (both from HCPs and service users’ perspective) will highlight important themes pertaining to the existing health literacy channels, gaps and challenges and strategies to further strengthen its implementation to promote patient centered care and further improve the quality of the services.

4. Data Analysis Plan

• Although HLQ scoring methods are referenced, the manuscript would benefit from a concise overview of intended statistical methods, including any subgroup analyses, correlation assessments, or use of regression models.

Response: Thank you for the important point. The HLQ scoring details are provided in text and tabular form now in the methods as kindly suggested.

5. Conceptual Framework (Figure 1)

• The figure is valuable but visually complex. A clearer flow with consistent formatting and labeled sections would improve readability for international readers unfamiliar with Qatar's healthcare system.

Response: Thank you for the important comment. We have added an additional section in the methods as study settings which clearly outlines the primary health care system in the state of Qatar as kindly mentioned.

Moreover, we have added an additional figure (figure 2) which depicts Interrelationship of health literacy levels, challenges and barriers and strategies to improve health literacy of service users that the study aims to address by triangulation of the qualitative and quantitative data collected from the study. We have added a additional section in the methods as study settings which clearly outlines the primary health care system in the state of Qatar.

6. Language and Style

• Minor grammatical and structural edits are recommended. For example:

o Original: “Improved compliance with treatment leading to effective management…” Suggested: “…which may improve treatment compliance and disease management.”

• Data Availability Statement: While the current version is compliant, consider stating your intention to deposit anonymized data post-study (e.g., in a public repository or institutional archive).

Response: Thank you for highlighting the minor grammatical and structural edits. We have thoroughly re-read the manuscript and made the necessary changes as kindly suggested. The anonymised transcripts and digital recordings will be stored securely on a server within the clinical research department of Primary Health Care Corporation. This information is now included in revised version (track changes).

Overall Recommendation: Minor Revisions

The study protocol is methodologically sound, ethically appropriate, and well-aligned with current health system needs. Minor adjustments related to demographic data collection, recruitment rationale, and clarity of analysis plan are needed. Once addressed, this work will make a meaningful contribution to the literature on health literacy and chronic disease management in the region.

Response: Thank you for the positive feedback. It is really encouraging. Your kind suggestions have strengthened the study protocol, and we are grateful.

Reviewer 3

This protocol addresses a timely and important research topic with methodological rigor. Addressing the minor clarifications below will further strengthen the manuscript and ensure its value to the scientific and public health communities.

Response: Thank you for the positive feedback, it is really encouraging. Your important suggestions are now incorporated in the protocol which adds relevant details and strengthened the protocol. We are truly grateful.

Comment: Response Rate and Potential Bias

While the protocol anticipates a very low response rate and compensates by increasing the invitation pool, it may be beneficial to elaborate further on strategies to maximize response rates (e.g., use of reminders, engagement incentives) and to discuss potential non-response bias in greater detail.

Response: Thank you for the important comment. We have added details (track changes) elaborating strategies to maximize the response rate as kindly suggested. The identifying information (names phone numbers and HC numbers) of the targeted sample will be extracted by PHCC BHI department. The short message service (SMS) will be sent with an approved survey invitation text to the targeted sample. A reminder will be sent after a week. The invitation text message will contain a link to the online questionnaire form using Microsoft office Forms.

Comment: Qualitative Sampling:

The use of convenience sampling for qualitative interviews is acknowledged as a limitation. Consider discussing how heterogeneity will be maximized (e.g., purposive selection based on demographic characteristics) to enhance the credibility of qualitative findings.

Response: Thank you for the important comment. We have added these details in the recruitment strategy for conducting interviews among service users and FGD with healthcare providers. While the protocol was under review, we also got the amendment approved to conduct focus group discission with health care providers (HCPs) working in PHCC. The amendment approval letter by institutional review board is also attached in the supplementary documents. This further strengthens the study as qualitative analysis (both from HCPs and service users’ perspective) will highlight important themes pertaining to the existing health literacy channels, gaps and challenges and strategies to further strengthen its implementation to promote patient centered care and further improve the quality of the services.

Comment: Patient and Public Involvement:

If applicable, clarify whether patients or members of the public were involved in the development of the research questions, survey tool adaptation, or interview guide, as this is increasingly encouraged in health research.

Response: Thank you for the important comment. A section in the methods is added as patient and public involvement.

---

## [Decision Letter · Decision Letter 1]

2 Jul 2025

PONE-D-24-46029R1Exploring health literacy pertaining to general wellbeing and chronic disease management among population registered within Primary Healthcare System: A Study protocol.PLOS ONE

Dear Dr. Syed,

Thank you for submitting your manuscript to PLOS ONE. After careful consideration, we feel that it has merit but does not fully meet PLOS ONE’s publication criteria as it currently stands. Therefore, we invite you to submit a revised version of the manuscript that addresses the points raised during the review process.

We look forward to receiving your revised manuscript.

Kind regards,

Hansani Madushika Abeywickrama, Ph.D.

Academic Editor

PLOS ONE

Journal Requirements:

Reviewers' comments:

Reviewer's Responses to Questions

**Comments to the Author**

1. Does the manuscript provide a valid rationale for the proposed study, with clearly identified and justified research questions?

Reviewer #2: Yes

Reviewer #3: Yes

2. Is the protocol technically sound and planned in a manner that will lead to a meaningful outcome and allow testing the stated hypotheses?

Reviewer #2: Yes

Reviewer #3: Yes

3. Is the methodology feasible and described in sufficient detail to allow the work to be replicable?

Reviewer #2: Yes

Reviewer #3: Yes

4. Have the authors described where all data underlying the findings will be made available when the study is complete?

Reviewer #2: Yes

Reviewer #3: Yes

5. Is the manuscript presented in an intelligible fashion and written in standard English?

Reviewer #2: Yes

Reviewer #3: Yes

6. Review Comments to the Author

You may also provide optional suggestions and comments to authors that they might find helpful in planning their study.

Reviewer #2: The revised manuscript demonstrates significant improvement over the initial version. It now includes expanded methodological clarity, better integration of supporting literature, ethical considerations, and enhanced structure, making it a promising contribution to the field of public health in Qatar and beyond. However, several minor but important improvements are still required for optimal clarity and quality.

Methodology Enhancements

• Ensure consistency of terminology: For example, “non-probability convenience sampling” and “purposive selection” are both mentioned—be clear on how diversity is achieved in each phase.

• Timelines are helpful but could benefit from more detail regarding data analysis phases.

Quantitative Analysis – Clarified but Could Improve Further

• A concise overview of planned statistical analyses is still needed.

o A subgroup analyses using ANOVA or t-tests and assesses associations using regression models to evaluate factors associated with health literacy scores.

o Consider mentioning whether weighting adjustments will be used to compensate for the stratified sample.

Qualitative Approach – Improved with FGDs

• The inclusion of themes in the topic guides is useful. Consider condensing the description of FGD logistics (e.g., room arrangements, consent) for brevity unless required by journal format.

Patient and Public Involvement (PPI)

• PPI is acknowledged, and a section is now added.

• However, more reflective depth is needed. Consider stating how PPI will be integrated in future research phases.

Reviewer #3: This revised protocol demonstrates clear improvements and addresses previous reviewer concerns effectively. The study has strong potential to contribute valuable insights to the literature on health literacy and its role in chronic disease prevention and management in primary care settings.

I recommend acceptance with minor revisions, primarily to clarify the integration of mixed methods data, enhance response rate strategies, and consider final editorial polishing.

I wish the authors success in conducting this important and timely study

Minor Clarifications and Recommendations

Data Integration Approach:

While the manuscript mentions the triangulation of quantitative and qualitative findings, it would benefit from further elaboration on how this integration will be operationalized (e.g., through matrix comparison, joint displays, or narrative synthesis).

Qualitative Sampling and Saturation:

The authors indicate that interviews will continue until data saturation is reached. To strengthen this, it is recommended to briefly describe how saturation will be monitored or defined during data collection.

Strategies to Improve Response Rates:

Given the very low expected response rate (2%), it may be useful to discuss any additional measures to enhance participation, such as incentives, follow-up reminders, or community engagement strategies.

Conceptual Frameworks (Figures 1 & 2):

The addition of Figure 2 is appreciated; however, the visual complexity of the frameworks may be a challenge for general readers. Simplifying the layout and labeling the flow components more clearly could improve accessibility.

Data Availability Statement:

The current statement indicates that data will be made available upon study completion. The authors may consider specifying the repository (e.g., institutional archive or open-access platform) where de-identified datasets will be deposited, in line with open science principles.

Language and Style:

While the manuscript is generally well-written, a final proofreading may help correct minor typographical inconsistencies and improve flow, particularly in longer paragraphs in the introduction and discussion sections.

7. PLOS authors have the option to publish the peer review history of their article (what does this mean? ). If published, this will include your full peer review and any attached files.

**Do you want your identity to be public for this peer review?** For information about this choice, including consent withdrawal, please see our Privacy Policy .

Reviewer #2: No

Reviewer #3: **Yes: ** Ananda Chandrasekara

---

## [Author Response · Author response to Decision Letter 2]

7 Jul 2025

Dear Editor,

We truly appreciate the invaluable feedback by yourself and the reviewers. Incorporating the kind input has strengthened the quality of the study protocol and made it more detailed and scientifically sound. All the comments have been carefully considered, and we have tried our best to address them.

Sincere Regards

Dr Muslim Abbas Syed

MBChB, MPH, MSc., MS., MCPS HPE, PhD

Response to individual comments

Reviewer #2: The revised manuscript demonstrates significant improvement over the initial version. It now includes expanded methodological clarity, better integration of supporting literature, ethical considerations, and enhanced structure, making it a promising contribution to the field of public health in Qatar and beyond. However, several minor but important improvements are still required for optimal clarity and quality.

Response: Thank you for the kind comments. We truly appreciate your kind feedback which has helped us add more detailed information to improve the scientific quality of the paper.

Methodology Enhancements

• Ensure consistency of terminology: For example, “non-probability convenience sampling” and “purposive selection” are both mentioned—be clear on how diversity is achieved in each phase.

Response: Thank you for the kind comment. We have now ensured that the consistency is maintained in the manuscript. We have mentioned that the healthcare professionals and service users will be recruited from the various primary healthcare centers, ensuring that the sample is representative of the general population even though a non-probability sampling technique is utilized for the qualitative data collection.

• Timelines are helpful but could benefit from more detail regarding data analysis phases.

Quantitative Analysis – Clarified but Could Improve Further

• A concise overview of planned statistical analyses is still needed.

o A subgroup analyses using ANOVA or t-tests and assesses associations using regression models to evaluate factors associated with health literacy scores.

o Consider mentioning whether weighting adjustments will be used to compensate for the stratified sample.

Response: Thank you for the important comments regarding the quantitative analysis. We have now included additional information on descriptive and inferential data analysis as well as included template tables highlighting the various variables with P-values that will be highlighted at sub-group level analysis.

Qualitative Approach – Improved with FGDs

• The inclusion of themes in the topic guides is useful. Consider condensing the description of FGD logistics (e.g., room arrangements, consent) for brevity unless required by journal format.

Response: Thank you for your kind comment. We have included some additional logistical information as kindly suggested. Since the protocol is written for wider readership specific logistical details were avoided. But we agree with your relevant point and have included this information now.

• PPI is acknowledged, and a section is now added.

• However, more reflective depth is needed. Consider stating how PPI will be integrated in future research phases.

Response: Thank you for the kind comment. We have now included some additional information on how PPI will be integrated in future research phases.

Reviewer #3:

This revised protocol demonstrates clear improvements and addresses previous reviewer concerns effectively. The study has strong potential to contribute valuable insights to the literature on health literacy and its role in chronic disease prevention and management in primary care settings. I recommend acceptance with minor revisions, primarily to clarify the integration of mixed methods data, enhance response rate strategies, and consider final editorial polishing. I wish the authors success in conducting this important and timely study.

Response: Thank you for your kind words of encouragement. We are truly grateful for your feedback, which has been constructive, accurate and strengthens the quality of the study protocol. We have carefully considered all your comments and tried our best to address them.

Minor Clarifications and Recommendations

Data Integration Approach:

While the manuscript mentions the triangulation of quantitative and qualitative findings, it would benefit from further elaboration on how this integration will be operationalized (e.g., through matrix comparison, joint displays, or narrative synthesis).

Response: Thank you for this important point. We have added details about narrative synthesis and have also described how the evidence based Socioecological Model (SEM) can be potentially utilized to interpret the qualitative findings (the factors affecting the health literacy of service users accessing primary healthcare services) at intra-personal, inter-personal, institutional, community and policy level. The findings of the study will populate the framework. The findings of the HLQ (survey) provide invaluable information at subgroup population level which feeds into the intrapersonal factors affecting the health literacy of service users accessing the primary health care services and can be also presented in detail as an individual publication resulting from this study project. Considering your kind suggestion we have included an additional brief paragraph explaining the triangulation of qualitative and quantitative data.

Qualitative Sampling and Saturation:

The authors indicate that interviews will continue until data saturation is reached. To strengthen this, it is recommended to briefly describe how saturation will be monitored or defined during data collection.

Response: Thank you for highlighting a very important point. It is now described that saturation is achieved in qualitative data collection when no new additional information is provided by the new participants (service users) or no new ideas and themes are discussed. A point where the same concepts are being discussed. This concept is established and well documented in qualitative research and now a reference is also provided.

Strategies to Improve Response Rates:

Given the very low expected response rate (2%), it may be useful to discuss any additional measures to enhance participation, such as incentives, follow-up reminders, or community engagement strategies.

Response: Thank you for the important point. The strategies to enhance participation are now expanded and further described as kindly suggested.

Conceptual Frameworks (Figures 1 & 2):

The addition of Figure 2 is appreciated; however, the visual complexity of the frameworks may be a challenge for general readers. Simplifying the layout and labeling the flow components more clearly could improve accessibility.

Response: Thank you for the important comment. We also included a simplified figure 1 and interlinked it with figure 2 for clarity. The conceptual framework was included as they will form the basis of reporting the key findings of the study for publication.

Data Availability Statement:

The current statement indicates that data will be made available upon study completion. The authors may consider specifying the repository (e.g., institutional archive or open-access platform) where de-identified datasets will be deposited, in line with open science principles.

Response: Thank you for the important point. We have now included this information.

Language and Style:

While the manuscript is generally well-written, a final proofreading may help correct minor typographical inconsistencies and improve flow, particularly in longer paragraphs in the introduction and discussion sections.

Response: Thank you for the important point. As kindly suggested, a final proof reading was done. Thank you for this important suggestion.

---

## [Decision Letter · Decision Letter 2]

22 Jul 2025

PONE-D-24-46029R2Exploring health literacy pertaining to general wellbeing and chronic disease management among population registered within Primary Healthcare System: A Study protocol.PLOS ONE

Dear Dr. Syed,

Thank you for submitting your manuscript to PLOS ONE. After careful consideration, we feel that it has merit but does not fully meet PLOS ONE’s publication criteria as it currently stands. Therefore, we invite you to submit a revised version of the manuscript that addresses the points raised during the review process.

We look forward to receiving your revised manuscript.

Kind regards,

Hansani Madushika Abeywickrama, Ph.D.

Academic Editor

PLOS ONE

Journal Requirements:

Additional Editor Comments :

Please respond to the comments raised by Reviewer 3.

Reviewers' comments:

Reviewer's Responses to Questions

**Comments to the Author**

1. Does the manuscript provide a valid rationale for the proposed study, with clearly identified and justified research questions?

Reviewer #2: Yes

Reviewer #3: Yes

2. Is the protocol technically sound and planned in a manner that will lead to a meaningful outcome and allow testing the stated hypotheses?

Reviewer #2: Yes

Reviewer #3: Yes

3. Is the methodology feasible and described in sufficient detail to allow the work to be replicable?

Reviewer #2: Yes

Reviewer #3: Yes

4. Have the authors described where all data underlying the findings will be made available when the study is complete?

Reviewer #2: Yes

Reviewer #3: Yes

5. Is the manuscript presented in an intelligible fashion and written in standard English?

Reviewer #2: Yes

Reviewer #3: Yes

6. Review Comments to the Author

You may also provide optional suggestions and comments to authors that they might find helpful in planning their study.

Reviewer #2: The revised version of the manuscript demonstrates significant improvement. The study is well-conceived, aligns with public health priorities, and is methodologically sound. The mixed methods approach is justified, and the conceptual framework is clearly laid out. The integration of both quantitative and qualitative strands and alignment with theoretical models (SEM, HLQ, COREQ, SRQR) is commendable.

Reviewer #3: The study addresses an important and under-explored aspect of public health, particularly in multi-ethnic populations and health systems facing rising rates of chronic disease.

Strengths of the Manuscript:

The rationale for the study is well established, highlighting the links between health literacy, healthcare access, compliance, and chronic disease outcomes. The research objectives are clearly stated and justified.

The use of a robust mixed-methods design, combining quantitative (HLQ survey) and qualitative (service user interviews) approaches, is appropriate for capturing both the breadth and depth of health literacy issues in this population.

The methodology is described in sufficient detail to allow for replication. The stratified random sampling, use of a validated tool in both English and Arabic, and consideration of non-response rates are notable strengths.

The protocol is generally well written, presented in clear and standard English, and is logically structured. Ethical approval has been obtained, and plans for data sharing comply with PLOS ONE requirements.

Points Requiring Attention and Suggestions:

Response Rate and Non-Response Bias:

The protocol anticipates a very low survey response rate (2%) and addresses this with oversampling. While this is pragmatic, the potential for non-response bias remains. Please consider describing additional strategies to encourage survey participation and discussing how non-response bias will be assessed or mitigated in the analysis.

Qualitative Sampling and Diversity:

The qualitative arm relies on convenience sampling, which could limit the diversity of perspectives. Please clarify any steps that will be taken to ensure heterogeneity among interviewees (e.g., sampling across different clinics, demographic quotas) and describe how data saturation will be monitored and determined.

Integration of Mixed-Methods Findings:

Although both quantitative and qualitative methods are detailed, the protocol would be strengthened by a brief description of how these datasets will be integrated during analysis and interpretation to provide a comprehensive understanding of health literacy in the target population.

Operational Details for Replicability:

While the methodology is overall well-described, providing additional information on interviewer training, data management and security (especially for online data and interview recordings), and specific qualitative analysis procedures (e.g., coding framework, thematic analysis steps) would enhance transparency and replicability.

Clarity and Consistency:

The manuscript is generally clear and well-written, but minor typographical and stylistic edits are recommended. Ensure consistent terminology (for example, “service users” vs. “patients”) throughout the document for clarity.

Data Availability:

The commitment to data sharing is appropriate for a protocol. Upon study completion, please ensure that all underlying data are made available in accordance with PLOS ONE’s data policy, with attention to participant privacy and ethical considerations.

Additional Suggestions:

Consider including a brief timeline for study implementation, recruitment, and anticipated completion in the Methods section.

Discuss any potential challenges related to technology or language barriers that may affect online survey participation, especially among older or lower-literacy participants, and any mitigation strategies.

Outline your plan for communicating key findings to participating clinics, health authorities, or the wider public, to ensure that study results are translated into practical improvements.

7. PLOS authors have the option to publish the peer review history of their article (what does this mean? ). If published, this will include your full peer review and any attached files.

**Do you want your identity to be public for this peer review?** For information about this choice, including consent withdrawal, please see our Privacy Policy .

Reviewer #2: No

Reviewer #3: **Yes: ** Prof. Ananda Chandrasekara, Professor in Nutrition and Dietetics, Wayamba University of Sri Lanka

---

## [Author Response · Author response to Decision Letter 3]

28 Jul 2025

Reviewer #3: The study addresses an important and under-explored aspect of public health, particularly in multi-ethnic populations and health systems facing rising rates of chronic disease.

Strengths of the Manuscript:

The rationale for the study is well established, highlighting the links between health literacy, healthcare access, compliance, and chronic disease outcomes. The research objectives are clearly stated and justified.

The use of a robust mixed-methods design, combining quantitative (HLQ survey) and qualitative (service user interviews) approaches, is appropriate for capturing both the breadth and depth of health literacy issues in this population.

The methodology is described in sufficient detail to allow for replication. The stratified random sampling, use of a validated tool in both English and Arabic, and consideration of non-response rates are notable strengths.

The protocol is generally well written, presented in clear and standard English, and is logically structured. Ethical approval has been obtained, and plans for data sharing comply with PLOS ONE requirements.

Response: Thank you for the positive feedback and your kind review (of the two previous occasions review 1 and 2) which has strengthened the study protocol scientifically. We are truly grateful and have carefully considered the points highlighted in your 3rd review and have highlighted all the points which were suggested. Thank you for your time and continued support.

We have elaborated on the suggested details and signposted for your kind review. Each comment is responded separately, and we have mentioned in which section of the protocol the comment is addressed in the track changes document (main manuscript).

Points Requiring Attention and Suggestions:

Response Rate and Non-Response Bias:

The protocol anticipates a very low survey response rate (2%) and addresses this with oversampling. While this is pragmatic, the potential for non-response bias remains. Please consider describing additional strategies to encourage survey participation and discussing how non-response bias will be assessed or mitigated in the analysis.

Response: Thank you for the important point. We have highlighted in the protocol that the registered users will be sent reminder SMS after two weeks to address potential non-response bias. Moreover, as kindly suggested, we have further described in the analysis component employing appropriate statistical techniques to address the non-response bias and amend expected deviations in the population representativeness by the actual sample. This is highlighted under the data analysis plan for HLQ questionnaire component of the protocol as shown below:

‘The sampling method used will enable valid subgroup analysis by ensuring an adequate sample size in each subgroup. However, any calculated summary measure for the population parameter will be biased by deviations from the sampling frame. This type of bias is addressed by using appropriate subgroup weights, Table 1. Sensitivity analysis (a type of quantitative bias analysis) may be considered to measure the magnitude of sample deviation from the planned sample’.

Qualitative Sampling and Diversity:

The qualitative arm relies on convenience sampling, which could limit the diversity of perspectives. Please clarify any steps that will be taken to ensure heterogeneity among interviewees (e.g., sampling across different clinics, demographic quotas) and describe how data saturation will be monitored and determined.

Response: Thank you for the important point. We have highlighted in the last paragraph of the discussion component (track changes and clean version of the manuscript):

‘Another limitation could be the non-probability convenience sampling strategy to recruit service users to conduct interviews to capture their perceptions regarding the different health literacy channels. To address this issue, we will conduct interviews in different primary care clinics (out of the 32 clinics spread across the state of Qatar) till saturation of data is achieved (a point where no new concepts, ideas or themes are emerging, a concept which is evidence based and we have provided the reference) and will aim to include equal representation of male and female participants considering their ethnic backgrounds. Moreover, the FGD will include a diverse cohort (multidisciplinary & multiethnic) of health care providers (HCPs) involved in providing primary care in PHCC which will establish that the findings are generalizable, and the sample is representative’.

Integration of Mixed-Methods Findings:

Although both quantitative and qualitative methods are detailed, the protocol would be strengthened by a brief description of how these datasets will be integrated during analysis and interpretation to provide a comprehensive understanding of health literacy in the target population.

Response: Thank you for the important point. The immediate and the long-term outcomes of the research (qualitative and quantitative findings) is depicted and outlined in figure 2 as kindly highlighted in your comment. As suggested, we have highlighted in the manuscript (first paragraph of discussion, neat and track changes version):

‘The mixed methods study aims to assess the existing health literacy channels operating within the primary care health system and to identify the existing gaps and challenges from key stakeholders’ perspectives including both the services users registered with PHCC and the health care professionals. The key objective of the study also includes determining the health literacy levels of service users at population level. The findings of the HLQ highlight health literacy levels at population level and its interrelationship with the sociodemographic indicators whereas the in-depth qualitative investigation highlights the various factors nested within the SEM in context to health literacy of service users accessing primary care services and identifying the potential gaps and challenges. These findings will be triangulated to design an evidence-based health literacy framework which can be utilized for service design and re-design to deliver optimal patient centered primary health care services within the country & modelled in similar health care settings geographically’.

Operational Details for Replicability:

While the methodology is overall well-described, providing additional information on interviewer training, data management and security (especially for online data and interview recordings), and specific qualitative analysis procedures (e.g., coding framework, thematic analysis steps) would enhance transparency and replicability.

Response: Thank you for the important point. The qualitative component will be carried out by principal investigator who is a medical doctor with PhD and has published several qualitative studies in peer reviewed Q1 journals. This will be highlighted in the COREQ & SRQR checklist. It is mentioned in the methods section that the study will be reported in accordance with these checklists. It is further highlighted in the manuscript:

‘The healthcare managers of the primary health centers affiliated with Primary Health Care Corporation will be contacted via email to obtain permission to approach service uses by accessing the service to participate in face-to-face interviews while they wait for their consultation in the waiting area of the health center. A separate room will be requested to be allocated by the health manager of the primary health center to conduct the interviews. The service users will be approached by the principal investigator (MA) and invited to participate in the study. When a service user agrees to participate, they will be invited to the allotted interview room. Prior to starting the interview, the purpose of the study will be explained in detail to the service user, and they will be given the opportunity to ask further questions to ensure that their participation is voluntary, informed and non-coercive. The written consent will be taken by the interviewer before starting the interview. The interviews will be recorded with permission of the service user. Once the interview has taken place all contact information about the service user will be securely destroyed and all the data collected will be anonymised. The interviews are semi-structured and will be conducted utilising an interview guide and script (supplementary document). The interviews will last between 45 to 60 minutes.

Analysis of the qualitative data collected from FGD and interviews:

The qualitative data once recorded will be transcribed verbatim and then analyzed using thematic analysis. This approach encompasses ‘interpreting, exploring, and reporting patterns and clusters of meaning within the given data’ and will be facilitated by reading and re-reading the transcripts for a full familiarization. This will be followed by application of open codes to four transcripts to identify emerging themes of relevance by two researchers (MA AND ASA). A Computer Assisted Qualitative Data Analysis (CAQDAS) package (NVivo 12 for Windows) will be utilized for this process. This will be followed by agreement by the two researchers (MA and ASA) on a set of codes which will be used with the rest of the transcripts. During this stage categories will be constructed and defined by grouping of codes. This will lead to the development of a working coding framework which will be utilized with the rest of the data and amended as necessary. The study will report the results in accordance with ‘Consolidation criteria for reporting qualitative research (COREQ) and ‘Standards for reporting qualitative research ‘(SRQR) guidelines.’

Clarity and Consistency:

The manuscript is generally clear and well-written, but minor typographical and stylistic edits are recommended. Ensure consistent terminology (for example, “service users” vs. “patients”) throughout the document for clarity.

Response: Thank you for the important point. The consistency has been now maintained in setting out the objectives in the introduction component as kindly suggested. The terms service users and patients had to be used interchangeably later in the text because in certain components of the protocol we had to highlight the term ‘service users’ to specifically refer to population accessing primary health care centers and registered with PHCC. And people living with chronic disease conditions had to be referred to as patients.

Data Availability:

The commitment to data sharing is appropriate for a protocol. Upon study completion, please ensure that all underlying data are made available in accordance with PLOS ONE’s data policy, with attention to participant privacy and ethical considerations.

Response:

Thank you for the important point. It is now highlighted in the ethical consideration section as kindly suggested:

‘Written consent will be obtained by the principal investigator (MA) prior to the start of the interviews with service users and will be recorded on the digital audio recorder with the participants’ permission. At all stages of the interview, it will be emphasised that participation is voluntary, and that the data generated will be anonymous. It will be ensured and made clear to the participants that they could withhold any information that they feel is too sensitive or withdraw from the study at any time. The interviews will be transcribed and anonymised soon after the interviews and the digital recordings of the interviews will be removed from the digital devices. The anonymised transcripts and digital recordings will be stored securely on a server within the clinical research department of Primary Health Care Corporation. Ethics approval was received to conduct the study from Institutional Review Board of Primary Health Care Corporation (supplementary document).

Additional Suggestions:

Consider including a brief timeline for study implementation, recruitment, and anticipated completion in the Methods section.

Response: Thank you for the important point. A timeline has been added in tabular form and highlighted as suggested.

---

## [Decision Letter · Decision Letter 3]

19 Aug 2025

PONE-D-24-46029R3Exploring health literacy pertaining to general wellbeing and chronic disease management among population registered within Primary Healthcare System: A Study protocol.PLOS ONE

Dear Dr. Syed,

Thank you for submitting your manuscript to PLOS ONE. After careful consideration, we feel that it has merit but does not fully meet PLOS ONE’s publication criteria as it currently stands. Therefore, we invite you to submit a revised version of the manuscript that addresses the points raised during the review process.

We look forward to receiving your revised manuscript.

Kind regards,

Hansani Madushika Abeywickrama, Ph.D.

Academic Editor

PLOS ONE

Journal Requirements:

Reviewers' comments:

Reviewer's Responses to Questions

**Comments to the Author**

1. Does the manuscript provide a valid rationale for the proposed study, with clearly identified and justified research questions?

Reviewer #3: Yes

2. Is the protocol technically sound and planned in a manner that will lead to a meaningful outcome and allow testing the stated hypotheses?

Reviewer #3: Yes

3. Is the methodology feasible and described in sufficient detail to allow the work to be replicable?

Reviewer #3: Yes

4. Have the authors described where all data underlying the findings will be made available when the study is complete?

Reviewer #3: Yes

5. Is the manuscript presented in an intelligible fashion and written in standard English?

Reviewer #3: Yes

6. Review Comments to the Author

You may also provide optional suggestions and comments to authors that they might find helpful in planning their study.

Reviewer #3: Thank you for the opportunity to review this well-constructed and timely study protocol addressing health literacy in primary healthcare settings in Qatar. The manuscript presents a clear rationale, an appropriate mixed-methods approach, and demonstrates careful consideration of both methodological rigor and ethical compliance. The following comments summarize the strengths of the protocol and provide suggestions for improvement to enhance clarity, replicability, and alignment with PLOS ONE’s expectations.

1. Rationale and Research Questions

The manuscript presents a strong rationale for the study, supported by a comprehensive review of the literature. The authors effectively highlight the significance of health literacy in the context of chronic disease management and general wellbeing, particularly in multicultural populations. The research questions and objectives are clearly defined, relevant, and appropriately justified. The use of the Socioecological Model (SEM) as a conceptual framework adds depth and relevance to the study design.

2. Methodological Soundness and Analytical Strategy

The mixed-methods approach—integrating the Health Literacy Questionnaire (HLQ) and qualitative interviews/focus groups—is well-justified and methodologically sound. The sampling strategy for the quantitative component is detailed and appropriate, with clear justification for oversampling due to the anticipated low response rate. The inclusion of subgroup weighting and sensitivity analysis is commendable and strengthens the analytical plan.

The qualitative component is described in detail, including participant recruitment, data collection procedures, thematic analysis approach using NVivo, and adherence to COREQ and SRQR guidelines. The triangulation of quantitative and qualitative findings is a major strength and is well-explained.

However, I recommend the following minor improvements:

Statistical power calculation: Although the sample size is large, inclusion of a formal power analysis for the HLQ component would further validate the sampling rationale.

Clarify exploratory elements: Consider clearly identifying which components of the analysis (particularly within the qualitative arm) are exploratory, and how these will be distinguished in reporting.

3. Feasibility and Replicability

The study appears feasible within the stated timeline and is supported by access to a large registered population within PHCC. Methodological descriptions, including sampling, recruitment, data management, and ethical procedures, are sufficiently detailed to allow for replication. The use of a validated instrument (HLQ) and the structured thematic analysis framework provide a solid foundation for reproducibility.

Suggestions:

A brief mention of interviewer training (e.g., for conducting interviews and FGDs) would enhance transparency.

Consider including a brief outline of data monitoring or audit mechanisms, if any, to strengthen quality assurance.

4. Data Availability

The authors appropriately address data availability in accordance with PLOS ONE’s policy. While no datasets are yet generated, the authors commit to making de-identified data available upon study completion, with proper ethical safeguards. This is acceptable at the protocol stage.

5. Language and Presentation

The manuscript is generally well-written and intelligible. The use of standard academic English is appropriate, and the document is logically organized. Minor typographical and stylistic issues exist (e.g., occasional inconsistent use of “patients” vs. “service users”), but these do not impede understanding. A final proofreading prior to acceptance is recommended to correct minor inconsistencies.

6. Ethical Considerations

The study demonstrates full compliance with ethical standards. IRB approval is clearly stated, informed consent procedures are appropriate, and participant confidentiality is well-addressed. The manuscript includes clear information about how data will be anonymized, stored, and handled securely.

7. Additional Suggestions (Optional)

The inclusion of figures and conceptual models (SEM and health literacy framework) adds value. Ensure these are high-resolution and clearly labeled in the final submission.

Consider adding a brief dissemination plan, outlining how findings will be shared with stakeholders (e.g., healthcare providers, policymakers, or the public) to maximize impact.

Given the diversity of the study population, translation validation procedures (for the HLQ and interview guides) could be described in more detail to ensure cultural and linguistic appropriateness.

7. PLOS authors have the option to publish the peer review history of their article (what does this mean? ). If published, this will include your full peer review and any attached files.

**Do you want your identity to be public for this peer review?** For information about this choice, including consent withdrawal, please see our Privacy Policy .

Reviewer #3: **Yes: ** Ananda Chandrasekara

---

## [Author Response · Author response to Decision Letter 4]

20 Aug 2025

Response to reviewer comments

Reviewer #3: Thank you for the opportunity to review this well-constructed and timely study protocol addressing health literacy in primary healthcare settings in Qatar. The manuscript presents a clear rationale, an appropriate mixed-methods approach, and demonstrates careful consideration of both methodological rigor and ethical compliance. The following comments summarize the strengths of the protocol and provide suggestions for improvement to enhance clarity, replicability, and alignment with PLOS ONE’s expectations.

Response: Thank you for the detailed review and useful suggestions in the previous reviews. We are grateful for your time and expert input. Your feedback has made the protocol very much detailed and scientifically robust. We have also considered your minor revision suggestions in this 4th round of review and have incorporated all the changes kindly suggested by you.

1. Rationale and Research Questions

The manuscript presents a strong rationale for the study, supported by a comprehensive review of the literature. The authors effectively highlight the significance of health literacy in the context of chronic disease management and general wellbeing, particularly in multicultural populations. The research questions and objectives are clearly defined, relevant, and appropriately justified. The use of the Socioecological Model (SEM) as a conceptual framework adds depth and relevance to the study design.

Response: Thank you for the kind comments. Your suggestions in the previous reviews were very helpful and strengthened the protocol rationale and highlighted the relevant frameworks (adding additional details). We are grateful.

2. Methodological Soundness and Analytical Strategy

The mixed-methods approach—integrating the Health Literacy Questionnaire (HLQ) and qualitative interviews/focus groups—is well-justified and methodologically sound. The sampling strategy for the quantitative component is detailed and appropriate, with clear justification for oversampling due to the anticipated low response rate. The inclusion of subgroup weighting and sensitivity analysis is commendable and strengthens the analytical plan.

Response: Thank you for the kind comments. The inclusion of subgroup weighting and sensitivity analysis was one of your kind suggestions in the 3rd round of review previously. Thank you for your feedback which has strengthened the protocol scientifically.

The qualitative components are described in detail, including participant recruitment, data collection procedures, thematic analysis approach using NVivo, and adherence to COREQ and SRQR guidelines. The triangulation of quantitative and qualitative findings is a major strength and is well-explained.

Response: Thank you for the kind comments and encouragement.

However, I recommend the following minor improvements:

Statistical power calculation: Although the sample size is large, inclusion of a formal power analysis for the HLQ component would further validate the sampling rationale.

Response: Thank you for the important comment. We have added the following details as kindly suggested in the sampling strategy section:

‘A sample size of only 2000 will detect a very small difference of 0.1 in the mean score of a specific domain (ranging between 1 and 4 with a standard deviation of 1) between two groups with an estimated Beta power of 0.99 at alpha of 0.05. Therefore, the targeted sample size of 6000 will detect virtually any difference or effect’.

Clarify exploratory elements: Consider clearly identifying which components of the analysis (particularly within the qualitative arm) are exploratory, and how these will be distinguished in reporting.

Response: Thank you for the comment. We have provided more details as kindly highlighted in your important point in the interpretation of qualitative findings (the FGD and interviews will feed the SEM model). It is now highlighted in that section as outlined below:

‘The qualitative analysis will be interpreted utilizing the Socioecological Model (SEM) which will provide to explore the wider determinants of health literacy of service users accessing the primary health care services. SEM is a widely accepted framework for comprehending and describing health determinants and in its entirety can be effectively utilized to explore the factors associated with health literacy of service users nested within the SEM. Although the SEM is widely acknowledged as a framework for understanding wider health determinants, there are only a few examples embedding health literacy in socioecological context. The existing literature only partially explores determinants nested within the SEM and does not investigate the model’s entirety when examining factors related to health literacy of service users accessing primary care services. The state of Qatar has a highly developed and extensive primary care health system, Primary Health Care Corporation (PHCC) which is composed of 32 primary health care centers which are scattered throughout the country. PHCC has the prime objective to deliver universal health coverage to its registered population. Each primary health care center is equipped with modern healthcare facilities managed by an international multidisciplinary team of health care professionals delivering the highest standards of primary health care services to a diverse multi-ethnic population (comprising of approximately 88% expats). However, studies report that despite the strong primary health care infrastructure, Qatar has a high prevalence of non-communicable disease and challenges associated with uptake of the services among the population registered with PHCC with diverse ethnic backgrounds. The model will be populated by the important themes emerging from the qualitative interviews with the patients and FGDs with HCWs to highlight the intrapersonal, interpersonal, Institutional, Community and Policy level factors as illustrated in figure 1.

3. Feasibility and Replicability

The study appears feasible within the stated timeline and is supported by access to a large, registered population within PHCC. Methodological descriptions, including sampling, recruitment, data management, and ethical procedures, are sufficiently detailed to allow for replication. The use of a validated instrument (HLQ) and the structured thematic analysis framework provide a solid foundation for reproducibility.

Response: Thank you for the kind comments. The previous reviews and kind suggestions have substantially strengthened the protocol.

Suggestions:

A brief mention of interviewer training (e.g., for conducting interviews and FGDs) would enhance transparency.

Response: Thank you for highlighting an important point. The interviews and FGD will be conducted by Principal investigator who is a medical doctor with multiple master’s degrees and PhD. He is a published researcher and has substantial experience of publishing qualitative research in Q1 high impact journals. This is also a criterion in the COREQ and SRQR checklist. We have added this information as highlighted in your kind comment.

‘The interviews will be conducted by PI. He has the relevant academic qualifications and substantial experience of conducting qualitative research and publishing in high impact journals. This is in accordance with the criterion of COREQ and SRQR regarding standards required by the researcher conducting qualitative research.’

Consider including a brief outline of data monitoring or audit mechanisms, if any, to strengthen quality assurance.

Response: Thank you for highlighting a very important point. We have added a paragraph on quality control and good practice measures:

‘To ensure the ethical and regulatory integrity of the study, oversight mechanisms will be implemented throughout its duration. A designated research monitor from the Clinical Research Department will conduct periodic monitoring and auditing activities. These will verify that the study is being conducted in accordance with the protocol approved by the Institutional Review Board (IRB), and in compliance with Good Clinical Practice (GCP) guidelines and Policies, Regulations and Guidelines for Research Involving Human as outlined by Ministry of Public Health (MoPH) Qatar. Monitoring will include the identification and documentation of any protocol deviations or instances of non-compliance. All findings will be addressed promptly to uphold participant safety and maintain the scientific validity of the research.’

4. Data Availability

The authors appropriately address data availability in accordance with PLOS ONE’s policy. While no datasets are yet generated, the authors commit to making de-identified data available upon study completion, with proper ethical safeguards. This is acceptable at the protocol stage.

Response: Thank you for the kind comments.

5. Language and Presentation

The manuscript is generally well-written and intelligible. The use of standard academic English is appropriate, and the document is logically organized. Minor typographical and stylistic issues exist (e.g., occasional inconsistent use of “patients” vs. “service users”), but these do not impede understanding. A final proofreading prior to acceptance is recommended to correct minor inconsistencies.

Response: Thank you for the kind comments. A final proof reading is now done.

6. Ethical Considerations

The study demonstrates full compliance with ethical standards. IRB approval is clearly stated, informed consent procedures are appropriate, and participant confidentiality is well-addressed. The manuscript includes clear information about how data will be anonymized, stored, and handled securely.

Response: Thank you for the kind comments. Your input was indeed very useful and has further strengthened the protocol.

7. Additional Suggestions (Optional)

The inclusion of figures and conceptual models (SEM and health literacy framework) adds value. Ensure these are high-resolution and clearly labeled in the final submission.

Consider adding a brief dissemination plan, outlining how findings will be shared with stakeholders (e.g., healthcare providers, policymakers, or the public) to maximize impact.

Response: the following section is added as dissemination plan as kindly suggested:

Dissemination Plan

Upon completion of the study, findings will be disseminated through multiple channels to ensure a broad and meaningful impact. Results will be submitted for publication in peer-reviewed journals and presented at relevant scientific conferences. In addition, tailored summaries will be shared with key stakeholders, including healthcare providers and policymakers, to inform practice and decision-making. Where appropriate, public-facing materials will be developed to communicate findings to the public, thereby enhancing transparency and promoting community engagement with the research outcomes.

---

## [Editor Report · Decision Letter 4]

11 Sep 2025

Exploring health literacy pertaining to general wellbeing and chronic disease management among population registered within Primary Healthcare System: A Study protocol.

PONE-D-24-46029R4

Dear Dr. Syed,

We’re pleased to inform you that your manuscript has been judged scientifically suitable for publication and will be formally accepted for publication once it meets all outstanding technical requirements.

Kind regards,

Hansani Madushika Abeywickrama, Ph.D.

Academic Editor

PLOS ONE
---

## [Editor Report · Acceptance letter]

PONE-D-24-46029R4

PLOS ONE

Dear Dr. Syed,

I'm pleased to inform you that your manuscript has been deemed suitable for publication in PLOS ONE. Congratulations! Your manuscript is now being handed over to our production team.

Kind regards,

on behalf of

Dr. Hansani Madushika Abeywickrama

Academic Editor

PLOS ONE